# Target Definition in MR-Guided Adaptive Radiotherapy for Head and Neck Cancer

**DOI:** 10.3390/cancers14123027

**Published:** 2022-06-20

**Authors:** Mischa de Ridder, Cornelis P. J. Raaijmakers, Frank A. Pameijer, Remco de Bree, Floris C. J. Reinders, Patricia A. H. Doornaert, Chris H. J. Terhaard, Marielle E. P. Philippens

**Affiliations:** 1Department of Radiotherapy, University Medical Center Utrecht, 3584 Utrecht, The Netherlands; c.p.j.raaijmakers@umcutrecht.nl (C.P.J.R.); f.c.j.reinders-2@umcutrecht.nl (F.C.J.R.); p.a.h.doornaert@umcutrecht.nl (P.A.H.D.); c.h.j.terhaard@umcutrecht.nl (C.H.J.T.); m.philippens@umcutrecht.nl (M.E.P.P.); 2Department of Radiology, University Medical Center Utrecht, 3584 Utrecht, The Netherlands; f.a.pameijer@umcutrecht.nl; 3Department of Head and Neck Surgical Oncology, University Medical Center Utrecht, 3584 Utrecht, The Netherlands; r.debree@umcutrecht.nl

**Keywords:** head and neck cancer, MR-guided radiotherapy, adaptive radiotherapy, MRI, diffusion-weighted imaging, IGRT, oropharyngeal cancer

## Abstract

**Simple Summary:**

Adaptive radiotherapy for head and neck cancer has become more routine due to an increase in imaging quality and improvement in radiation techniques. With the availability of faster adaptive workflows, it is possible to adapt more easily to (daily) changes. MRI offers besides great anatomical imaging, also functional information about the tumor and surrounding tissue. The aim of this review is to provide current state of evidence about target definition on MRI for adaptive strategies in the treatment of head and neck cancer.

**Abstract:**

In recent years, MRI-guided radiotherapy (MRgRT) has taken an increasingly important position in image-guided radiotherapy (IGRT). Magnetic resonance imaging (MRI) offers superior soft tissue contrast in anatomical imaging compared to computed tomography (CT), but also provides functional and dynamic information with selected sequences. Due to these benefits, in current clinical practice, MRI is already used for target delineation and response assessment in patients with head and neck squamous cell carcinoma (HNSCC). Because of the close proximity of target areas and radiosensitive organs at risk (OARs) during HNSCC treatment, MRgRT could provide a more accurate treatment in which OARs receive less radiation dose. With the introduction of several new radiotherapy techniques (i.e., adaptive MRgRT, proton therapy, adaptive cone beam computed tomography (CBCT) RT, (daily) adaptive radiotherapy ensures radiation dose is accurately delivered to the target areas. With the integration of a daily adaptive workflow, interfraction changes have become visible, which allows regular and fast adaptation of target areas. In proton therapy, adaptation is even more important in order to obtain high quality dosimetry, due to its susceptibility for density differences in relation to the range uncertainty of the protons. The question is which adaptations during radiotherapy treatment are oncology safe and at the same time provide better sparing of OARs. For an optimal use of all these new tools there is an urgent need for an update of the target definitions in case of adaptive treatment for HNSCC. This review will provide current state of evidence regarding adaptive target definition using MR during radiotherapy for HNSCC. Additionally, future perspectives for adaptive MR-guided radiotherapy will be discussed.

## 1. Introduction

Head and neck squamous cell carcinoma (HNSCC) is the sixth most common form of cancer in humans and accounts for about 4% of all cancers arising in Europe [1]. Classic etiological factors for head and neck cancer are (ab)use of nicotine and alcohol [2]. A subset of oropharyngeal cancer is associated with a Human Papilloma Virus (HPV) infection [3]. Currently, HPV-associated tumors accounts for about 40–50% of all oropharyngeal cancers. Patients with HPV-related tumors present themselves more often with higher stage disease, and treatment response is different. HPV-related tumors generally respond better, but there is a discrepancy between primary tumors and lymph node metastases. The primary tumors generally respond quicker, where the nodal respond is delayed and can take up to 6 months before they completely respond [4].

The overall survival of HNSCC patients with oropharyngeal cancer especially has improved in recent decades. Explanations for that are the increase in incidence of HPV-related tumors, improved staging due to improvements in imaging and improved treatment modalities [1]. For HPV-unrelated tumors (e.g., laryngeal carcinoma), there is no increase found in overall survival.

Since the introduction of intensity-modulated radiotherapy (IMRT), organ preserving treatment with (chemo)radiotherapy has become mainstay in treatment of HNSCC. IMRT led to more conformity in target coverage while lowering the dose to organs at risk (OAR) and thereby a significant reduction in late toxicity [5,6,7]. Nonetheless, late toxicity (i.e., xerostomia and dysphagia) is still a big problem and has a pivotal role in quality of life for HNSCC survivors [8]. In particular, since the survival of (HPV-related) HNSCC patients is increasing, late toxicity of treatment becomes more important. Radiation delivery needs to be improved in order to limit late toxicity even more. Most crucial is the delineation of the tumor and OARs. Although delineation of the OARs is standardized and, in general, conformity index is acceptable in studies [9,10], delineation of the tumor, and less prominent delineation of the neck nodes may show more interobserver variation among radiation oncologists [11]. Improvement in validated imaging techniques is mandatory. Secondly, an improvement of radiation techniques may further increase the therapeutic window (limit the dose to OARs while maintaining an optimal dose to the targets).

The main focus of image-guided radiotherapy (IGRT) is to broaden this therapeutic window. With all the improvements in imaging, such as increased resolution of the 2-deoxy-2-[fluorine-18]fluoro-D-glucose positron emission tomography with computed tomography (18F-FDG-PET/CT), multiparametric magnetic resonance imaging (mpMRI) with diffusion-weighted imaging (DWI) and dynamic contrast-enhanced (DCE) MRI, the landscape of target definition has changed drastically in comparison to the contrast-enhanced computed tomography (ce-CT)-only workflow [12].

With the introduction of new radiotherapy techniques (i.e., integrated MR-guided EBRT, proton/heavy ion therapy, CBCT-adaptive EBRT), IGRT entered a new chapter in the IGRT era. With the integration of MRI, interfraction and intrafraction changes have become visible. An integrated daily adaptive workflow (Figure 1) allows regular and fast adaptation of target areas. In proton therapy, adaptation is even more important in order to obtain high quality dosimetry. The changing air cavities (air or fluid) in the larynx or paranasal sinuses are of big influence of the Bragg peak locations and thereby the dose distribution. This effect is more outspoken in proton therapy than it is in photon therapy [13,14]. The field of integrated MR-guided proton therapy is also moving forward [15]; nonetheless, it will take several years for this technique to become clinically available. This topic will not be described in this review. Additionally, proton therapy-specific adaptive strategies or possibilities, such as in vivo dosimetry using prompt gamma measurements [16], will not be discussed in this review.

The question is which adaptations during radiotherapy treatment are oncology safe and at the same time provide better sparing of OARs in case of treatment of HNSCC.

This review will provide the current state of evidence regarding adaptive target definition using MR during radiotherapy for HNSCC and will provide future perspectives for adaptive MR-guided radiotherapy.

## 2. Tumor Target Definition before RT Treatment

### 2.1. The Current Role of CT

A CT scan in treatment position (i.e., immobilized on a flat table top) is currently essential for photon-based radiotherapy. The electron density information acquired from CT is used for patient-specific dose calculation in treatment planning. For proton therapy planning, CT is also used to calculate the stopping power. Another advantage of using a CT scan in treatment position is the geometrical accuracy of the image. Relative disadvantages of using CT in head and neck cancer patients are poor soft tissue contrast and susceptibility for metal artefacts of dental fillings [17]. Artefacts can be reduced by using metal artefact reduction techniques; nonetheless, they may cause serious difficulties in target delineation and, moreover, difficulties in dose calculation. All currently available delineation consensus guidelines of both tumor and elective lymph node levels are CT-based [18,19]. Despite the international consensus guidelines, there still is significant interobserver variation in delineation of target volumes. Distances between CTV contours of different observers of more than 20 mm are regularly observed [11,20,21].

### 2.2. The Current Role of Multiparametric MRI

#### Anatomical Imaging

In comparison to CT, MRI provides superior soft tissue contrast that improves delineation of tumor and organs at risk [22]. For modern head and neck cancer radiotherapy multiparametric MRI (T1, T1gd with and without fat suppression, possible when using Dixon techniques [23], T2 and DWI) is required. The current workflow requires that the MRI in treatment position (i.e., thermoplastic mold) is co-registered with the CT to facilitate delineation of the gross tumor volume (GTV) on CT scan. The use of MRI that is co-registered with the CT images significantly reduces interobserver variability regarding GTV delineation in nasopharyngeal cancer [24,25]. For delineation of primary tumors at other HN subsites the contribution of multimodal imaging is less clear [26,27,28]. Overall, the mean GTV volume increases and interobserver variation decreases with the addition of MRI and PET/CT. However, delineation of GTV based on MR only showed a wide variation in a study with 4 oropharyngeal cancer patients and 24 observers [29]. Guidelines for the delineation using MRI are of utmost importance.

However, the problem is that these series compare delineations of experts (radiologists and radiation oncologists), but a gold standard (such as histopathology) is missing. For the larynx, there are some series that correlated (multimodal) imaging to histopathology [30,31,32] showing that accurate GTV delineation is not easy, even with use of all modern imaging techniques. On conventional sequences (T1, T1gd FS and T2), even with the use of specific MR criteria [33] delineation of the tumor versus surrounding tissues remained quite variable between different experts. DWI seems to create a clearer boundary between tumor and surrounding tissue. However, conventional EPI DWI has an inherently poor geometric accuracy [34]. Non-EPI techniques, such as Splice [34], may improve delineation (see below).

### 2.3. The New Role of Multiparametric MRI

#### 2.3.1. Anatomical Imaging

The advances in head and neck imaging over recent decades has forced the radiation oncology community to reconsider our target definitions, as has been proposed by Kaanders et al. in their opinion paper [12].

As image quality has improved over recent decades with the upcoming use of MRI, the tumor outline on imaging has become better visible and by that the GTV could potentially be better defined. The problem is that with MRI, the tissue is visualized in much more detail than CT or PET-CT, so there is a need for guidelines that state which MR sequences are the standard and which signal intensities to delineate. Consequently, the clinical target volume (CTV) margin of 10 mm or larger, as generally applied, might be reduced, as this accounts for tumor depositions or extensions that are not visualized by the imaging technique used [19]. Nonetheless, CTV margin also seems to be used to compensate for interobserver delineation variation [35].

So, the question is if this margin is still necessary. Several series already showed that margin reduction either by switching from contouring the whole oropharynx to using a CTV margin of 10 mm [36] or by decreasing the GTV-CTV-PTV margin with 6 mm [37,38] results in significant reduction in acute and long term toxicity without jeopardizing oncological outcome. Specifically, xerostomia ≥ grade II and dysphagia grade III are significantly reduced.

A CT-based atlas is used for delineation of organs at risk (OARs) [39]. Although the atlas caused fairly good conformity in the delineation of parotid glands, submandibular glands and spinal cord, there still remains variability in CT-based delineation of the pharyngeal constrictor muscles and oral cavity [10]. In particular, the radiation dose to the pharyngeal constrictor muscles, oral cavity and salivary glands play a crucial role in late radiation-induced toxicity [8,40,41], so accurate delineation (on MRI) is essential [42,43]. The prospective R-IDEAL stage 0 study, performed by the MR-Linac consortium, evaluated the inter- and intraobserver variation of target volume delineation on MRI [29]. Four patients with pre-treatment T1, T1 with gadolinium and T2 imaging were delineated by twenty-six radiation oncologists from seven international cancer centers. They were all experienced in both treating head and neck cancer patients. There was a large variability in GTV and CTV delineations of the primary tumor with dice similarity coefficients (DSC) varying in the range 0.69–0.77. The DSC of the OAR delineations of the different experts was above 0.8, showing that the OAR delineation on MRI is quite consistent.

#### 2.3.2. Functional Imaging

Besides all anatomical imaging, there is an immense potential in functional MR imaging for head and neck cancer. Functional imaging has the potential to identify heterogeneity within the tumor that may be interesting for dose escalation (resistant parts) or dose de-escalation (sensitive parts). Hypoxia of the tumor is considered a major cause of radio-resistance in head and neck cancer and decreased curability rates [44,45]. Hypoxia measurements on MRI are not possible, but perfusion measurement are. Perfusion of the tumor is linked to oxygenation of the tumor and can serve as surrogate marker for hypoxia. The complicating factor in this respect is that better-perfused tumors are better oxygenated, but hypoxia itself stimulates neo-angiogenesis, which also increases perfusion without improvement of oxygenation [46]. So, the role of perfusion as a surrogate marker for hypoxia is not clear yet.

*Dynamic contrast-enhanced (DCE)* MRI provides non-invasive in vivo evaluation of perfusion and permeability of blood vessels. These factors can be seen as surrogate marker for hypoxia within malignant tumor tissue [47], with the above mentioned precaution of overestimation due to neo-angiogenesis. The improved spatial and temporal resolution of MR scanners allows us to evaluate the distribution of contrast in vivo. This results in tissue-unique distribution curves regarding permeability and perfusion. DCE-MRI studies have shown an association between tumors with high perfusion signals and improved oncological outcomes [48,49,50,51,52,53,54,55,56]. A group from Michigan found that DCE-MRI was able to identify upfront subvolumes within primary tumors that corresponded with local failure [48]. These were subvolumes with significantly lower blood volume and blood flow compared to their surroundings. In broader perspective, hypoxia targeted dose painting dose escalation studies are developed and running [57,58,59]. Most of these studies use (18)F-FMISO PET to determine the hypoxic parts of the tumor and dose paint those parts up to 84 Gy. First results showed feasibility of such an approach, and oncological outcomes are still to be awaited. The problems in (18)F-FMISO are the availability and costs. Additionally, most PET images are static and made several hours after injection of (18)F-FMISO. There are also protocols in which dynamic distribution of fMISO is measured, for instance, at 1, 2 and 4 h with or without dynamic scanning for the first 15 min [60]. However, the radioactive decay of the tracer over time comes with the challenge of low activity and consequently low signal intensity. Therefore, besides the use of radioactivity, there is a need for further research to improve DCE MRI for this application.

*Blood or tissue oxygen level-dependent (BOLD or TOLD) imaging* provides information on regional differences in oxygenation. Changes in venous blood oxygenation is reflected in the T2*-weighted scans (BOLD). Additionally, changes in carbon dioxide tensions over time affects the BOLD signal due to the vascular reaction resulting in changes in blood volume. This principle was tested in patients that inhale increased levels of carbon-dioxide while undergoing MRI scanning with T2* imaging. Simultaneously, tissue oxygenation is measured by changes in T1 relaxation due to oxygen dissolved in tissue (TOLD) [61]. This allows for a noninvasive in vivo investigation of blood oxygenation and vascularity. These parameters can be seen as a surrogate marker for hypoxia. This effect has been tested for head and neck tumors [62]. The problems with BOLD imaging in the neck are the sensitivity of the coils, the spatial resolution, the sensitivity for magnetic field inhomogeneity and the geometric accuracy of the imaging. Small tumors or small lymph node metastases are not reliably represented. Currently, a study on the role of BOLD imaging as biomarker is enrolling patients with advanced tumors, and results are to be awaited [63].

*Diffusion-weighted imaging (DWI)* provides information on the microscopic mobility of water molecules, reflecting micro anatomical variations [64]. As the restriction of diffusional motion of water is greater in tumors, which have a high cell density, than in normal tissue, the differentiation between the normal tissue and tumor results in high-contrast images which are excellent for use in tumor discrimination [65]. A potential problem with using DWI in radiotherapy target delineation was the geometrical accuracy. A different acquisition with a geometrically accurate turbo spin-echo sequence covers this problem and makes it possible to use DWI in GTV delineation [34]. The accuracy of DWI in GTV delineation appears promising. A case series of three laryngeal cancer patients where DWI imaging was compared to the ground truth (i.e., histopathology) showed a superior conformity of GTV delineation based on DWI (GTV_dwi_) compared to GTV delineation based on clinical imaging [66]. As mentioned, this case series consisted only of three patients, and more mature data are to be awaited. Another major benefit of the high-contrast images of DWI is the opportunity to diminish interobserver variation due to clear margins, and also to optimize automatic contouring of GTV in order to both decrease variability and save time. The latter is especially important in view of daily adaptive MR-guided radiotherapy on the MR-Linac.

### 2.4. MR-Only Workflow

An MR-only workflow overcomes challenges posed when registering MR to CT images. The biggest problem in that is the difference in mobile anatomy due to tongue, tongue base and or larynx position. Additionally, swelling, swallowing or tumor progression can have an effect. This results in an uncertainty of about 2 mm in the center of the target position [67]. An MR-only workflow, in which a pseudo-CT is acquired, solves the registration issues and makes the simulation workflow more efficient. Voxel-based [68] or atlas-based methods [69] can be used to derive electron density maps from MR images, but a more recent method applies a convolutional neural network [70,71]. Generally, Dixon MR images are applied to separate water and fat, and either T1- or T2-weighted MRI is used. As bones do not have a signal intensity on MRI, this is the most challenging part. In particular, when treatment is delivered on a conventional linear accelerator with kV of CBCT, the quality of bone is crucial for proper matching and position verification. In addition, the derivation of air cavities as found in head and neck MRI is difficult. Ultimately, dose deviations of less than 1% can be obtained [70]. For now, the use of an MR-only workflow seems beneficial for delivery on an MR-Linac only.

### 2.5. MRI Scan Duration

A disadvantage of using MRI is the acquisition time. Scanning protocols with a duration of 30–45 min are common and every new sequence adds acquisition time or another sequence has to be replaced. Table 1 gives an overview of a clinically used scanning protocol for MRI in head and neck cancer patients. Added acquisition time, potentially increase the patients’ burden. Parallel imaging techniques are developed to accelerate MRI acquisition. Current areas of interest are compressed sensing techniques and artificial intelligence-based reconstructions to further minimize acquisition times. These techniques are expected to reduce the acquisition time by half [72].

## 3. Adaptive Tumor Target Definition during RT Treatment

### 3.1. Adaptation of Target Areas

Throughout the course of radiotherapy, the patient’s anatomy changes. This can alter the dose distribution and impact both target coverage as well as dose to organs at risk. Two main factors can be distinguished that could play a major role in dose distribution alteration during the course of radiotherapy, knowing the tumor regression/progression and functional changes in target areas.

#### 3.1.1. Adaptation Based on Tumor Regression/Progression

The first factor is the response of the tumor to the given treatment, both regression and progression. Weekly adaptation during treatment regarding volumetric changes is already a known concept from CT-based radiotherapy. A prospective study performed in the United Kingdom in 20 HNSCC patients evaluated the dosimetric consequences of a weekly adaptive strategy. They found a significant shrinkage of the primary tumor, but also a shrinkage and medial migration of the parotid glands [73]. This finding was confirmed by a recent study from van Timmeren et al. in 12 HNSCC patients treated on an MR-Linac [74]. They found, on average, a shrinkage of 30% in both submandibular and parotid glands during the treatment. This adaptive radiotherapy (ART) is meant to decrease the radiation dose to normal tissue and limit acute and late toxicities of radiotherapy. An ongoing study in the UMC Utrecht, in the Netherlands, evaluates treatment response of head and neck cancer patients during treatment using MRI. Patients undergo DWI and T2-weighted imaging prior to and during week 2, 3, 4 and 5 of the radiotherapy treatment and three weeks after radiotherapy. Volume changes are assessed on T2-weighted images and ADC changes are appreciated on the DWI images. The first results of 20 patients were presented at ESTRO 38 [75]. It was shown that, on average, tumors had shrunk more than 50% at the end of week 3 and over 80% at the end of week 5 (Figure 2). After week 4, the T2 images were hard to interpret and differentiation between malignant and nonmalignant tissue was difficult. In one patient, tumor progression was seen during treatment, and after treatment, this patient had residual disease. The value of ADC changes during treatment was not conclusive from this first analysis, due to the limited number of patients and recurrences.

#### 3.1.2. Adaptation Based on Functional Imaging of the Target Areas

The second factor is the adaptation of treatment based on functional imaging. The PREDICT-HN study evaluated the use of serial imaging during treatment in 41 HNSCC patients. This study showed that ADC values increased significantly during the course of radiotherapy. Interestingly, they found a discrepancy in ADC changes between primary and nodal sites [76]. Another study in 57 HNSCC patients [77] combined 18F-FDG-PET/CT and MRI (DWI, DCE, T2-weighted) imaging prior to and after 10 days of start of treatment. This study focused on prediction of outcome based on early imaging, but also provides important input for treatment adaptation. They showed that functional intratumor changes are predictive for loco-regional control. Two specific features (low delta ADC skewness and increased delta-*f*) appeared associated with loco-regional control. These findings are in line with previously studies in smaller HNSCC patient groups [49,78,79]. Whether these features hold their predictive value in serial imaging need to be confirmed in future studies. If so, this could provide opportunities for adaptation based on dynamic changes. All the above-mentioned modern imaging modalities, but mainly the development of the MR-Linac, demands more than just ART with homogeneous doses. Dose painting, i.e., delivery of inhomogeneous doses, or redistribution of dose is a concept described already in 2011 [80]. Due to the redistribution, dose escalation is possible on relatively resistant parts of the tumor without increasing the dose to entire PTV. Studies evaluating dose-painting, such as the EU ARTFORCE trial [81], all use static PET imaging, whereas it is known, for instance, that hypoxia changes from day to day, even from hour to hour. Functional imaging makes it possible to visualize day-to-day alterations when patients are being treated on the MR-Linac. As described above, subvolumes of the tumor that predict local failure can be defined on functional MRI. Nonetheless, it is not clear in the literature what dose needs to be painted to which (MR) features. There are technical possibilities, but there is an urgent need for more understanding and clear research on how imaging features translate to dose adaptations.

### 3.2. Motion Management of Target Areas

Motion management for HNSCC has generally been ignored in the past, because all patients are being treated in immobilization masks. Swallowing motion is observed, but it occurs rarely and does not need to be incorporated in the PTV margin. [82,83]. Specific treatments, such as single vocal cord irradiation [84], used 4D CT with respiratory phase rebinning for simulation. However, most institutions discount for the limited motion in PTV margins that generally amount to 3–5 mm [36,83] in regions where limited motion is observed. In order to be able to further reduce PTV margins for head and neck cancer, it is necessary to quantify and track motion during the fraction.

Motion quantification can be performed in several ways:Pre-treatment motion quantification for margin estimation. This method is used most often and is available in most modern radiotherapy departments. This can be performed using 2D cine images with a high frequency [85]. Another method is retrospectively rebinned 4D CT of 4D MRI to show 3D CTs or 3D MRIs of respiratory phases [86]. PTV margins incorporating this breathing related motion are extended by approximately 2 mm in cranio-caudal direction for laryngeal/hypopharyngeal tumors [83].Daily motion assessment before treatment starts using 2D cine MRI. A cine-MRI can be acquired as a part of the standard protocol and consists of continuous real-time imaging for a fixed time. On MR-guided systems, it is possible to acquire real-time cine MRIs to track the anatomy or tumor during treatment. This enables the use of patient’s individual PTV margins [83].Next step is online gating: real-time tracking using continuous 2D cines to track the position of the tumor and use gated delivery of radiation dose to limit margins and thereby limit the dose to the surrounding tissue. This method is mostly used in stereotactic radiotherapy of lung tumors [87]. The downside of this method is the increase in delivery time, which makes it less efficient.The ultimate step is online guidance or tracking: by trailing or tracking of the tumor using real-time cine images, the margins can be minimized with minimal dose to surrounding tissues. At this moment, imaging is not fast enough to perform online 3D tracking and trailing, but it will be in the next decade.

Head and neck cancer treatment is currently at step 2, with daily motion assessment using 2D cine MRI. The question is whether it is beneficial to use gating for HNSCC, because of the increased delivery time, or only for selected sub sites such as the mobile tongue (Figure 3). It seems more logical to wait for the technological innovations in tracking and trailing, since that would open up new possibilities for MR-guided HNSCC treatments with only limited margins.

## 4. Future Perspectives on Target Definition for HNSCC Treatments

### 4.1. New MR-L Treatments

The biggest issue in adaptively treating head and neck cancer patients on the MR-Linac is time. Both time on the machine as well as duration of the entire treatment course (i.e., 7 weeks) are long and intense for patients. The first, time on the machine, consists of scanning time, delineation, re-planning and delivery of the dose. In the current adapt-to-shape (ATS) workflow, the estimated total treatment time on the machine is between 45 and 60 min. For treatments in which there is only adapt to position (ATP), treatment times of 30 min are feasible. The delineation step on the machine will take at least 5–10 min and the plan adaptation phase another 5–10 min. There is an urgent need for workflow improvements, mainly aimed at reducing time spent on the machine. Human-centered artificial intelligence (AI) solutions are needed. An adaptive workflow consists of multiple steps where human interaction is needed. With the emerge of AI solutions, the role of the clinician shifts more towards quality assessment (QA). The time needed for QA of AI generated contours or treatment plans needs to be reduced in order to speed up the process. Inaccurate delineation could lead to decreased treatment quality (higher OAR dose) or even a worse oncological outcome of patients. However, not all inaccurate delineations lead to loss of quality, so there is need for proper error detection and prioritization. Regarding error detection, multiple solutions such as comparison of shape [88,89] or image characteristics are being proposed [90]. Essential is the visualization of the errors to the clinicians by automatically detect accurate from inaccurate regions [91,92] and present them as an overlay of “uncertainty isolines” [93]. The future challenge is to make the human-centered AI solutions reliable enough for clinicians to thrust the discrimination and thereby gaining speed in the process. Key input for these human-centered AI algorithms are well delineated volumes conform validated guidelines.

### 4.2. New Elective Nodal Target Definition

Conventional elective nodal irradiation has been proven effective in reducing the regional recurrence rate [94,95]. An international group of experts published consensus guidelines on the delineation of elective cervical lymph node levels [18]. These guidelines are meant to diminish practice variation and provide a standard for conducting clinical trials. The guidelines are based on anatomical boundaries visible on CT and on borders of the surgical anatomy. With the current strategy of targeting the fatty tissue in the neck, harboring the cervical lymph nodes, there is a regional recurrence rate of about 2–4% [96]. The elective neck node levels are based on the same levels that are removed by the head and neck surgeon. Apart from the lymph nodes, normal tissue is removed. Thus, elective lymph node-level irradiation comes at the price of increased normal tissue toxicity, and the questions arises whether it is necessary to target the entire fatty tissue and which radiation dose should be applied.

### 4.3. Treatment Adaptation of the Elective Lymph Node Regions

Treatment adaptation of the elective lymph node regions can be approached from two sides: reduction in dose and/or reduction in target volumes.

#### 4.3.1. Reducing the Dose

The discussion on lowering the dose given to elective lymph nodes is already ongoing. Elective radiation dose de-escalation studies have already been conducted [97,98,99,100] and definitive results are to be awaited. However, preliminary results appear promising regarding recurrence rate and lowering acute toxicity. Further discussion of dose reductions falls out of scope of this review.

#### 4.3.2. Reducing the Target Volume

A discussion that deserves more attention is about tailoring elective lymph node levels based on primary tumor site and/or individual lymph drainage pattern, especially in the current era with high-end multimodal imaging.

A study from the Netherlands Cancer Institute showed in 50 patients with lateralized HNSCCs that SPECT/CT-guided unilateral elective lymph node irradiation is safe regarding regional failures [101]. In this single-arm, phase II study, the contralateral neck was irradiated only if contralateral lymphatic drainage was observed, and only one contralateral recurrence was found during follow-up. At the same time, grade 2 and 3 toxicity significantly declined compared to a matched-pair group. Additionally, quality of life significantly improved after SPECT-guided elective lymph node radiotherapy [101]. The next study from that group, currently enrolling patients, is to include a sentinel lymph node biopsy in this SPECT/CT-guided radiotherapy. The contralateral neck is irradiated only if contralateral sentinel node is positive on histopathological examination [102]. The aim is to minimize unnecessary bilateral elective lymph node irradiation even further. The downside of this strategy is that an extra procedure (sentinel lymph node biopsy) is necessary.

The next step in elective lymph node irradiation is a new concept of individual lymph node irradiation, which was introduced in 2021 [103]. The idea of this concept is to target all individual lymph nodes visible on T2-weighted MR images (Figure 4). A planning study in laryngeal cancer patients showed that such a strategy results in a reduction in mean dose values to OARs up to 20 Gy [103]. A prerequisite for the implementation of this concept is having access to an MR-guided treatment modality, as small individual lymph nodes are not visible on cone beam CT, and day-to-day variation due to motion and shifts in individual lymph nodes needs to be corrected for. A prospective clinical trial will open up soon and begin enrolling patients. The aim of the study will be to test the feasibility of MR-Linac treatment in HNSCC patients.

In the head and neck area, the sentinel lymph node procedure is mostly applied in oral cancer [104]. To improve the sentinel lymph node procedure in early stage oral cancer, several new techniques have been developed [105]. A novel technology, currently being applied in sentinel lymph node research, is the use of interstitial MR lymphography [106,107]. After peritumoral injection of gadolinium or superparamagnetic iron oxide (SPIO), it is possible to visualize the lymphatic drainage of the primary tumor. Current research focuses on the possibility of finding sentinel lymph nodes with this technique [108,109]. The next step will be to validate the sentinel lymph node procedure for other HNSCC subsites, but it will also be to further develop the MR lymphography so that it is possible to not only identify the sentinel lymph node, but also predict whether this lymph node harbors metastases. With MR-guided systems, it is already possible to target individual lymph nodes [103], so the next step will be to only target draining lymph nodes on MR-guided systems and limit toxicity.

## 5. Conclusions

The use of MRI has become the standard of care in the treatment of head and neck cancer, and for therapeutic purposes especially, the role of MRI is emerging. With the increase in accuracy of radiotherapy delivery, MR-guided radiotherapy is the new chapter in radiotherapy. With the help of MRI guidance, both target volume reduction and dose de-escalation are being developed for primary and elective target areas. The possibilities of MRI in the radiotherapy guidance are immense, and novelties are arriving the coming years. This warrants specific international guidelines for target definition in MR-guided radiotherapy.

## Figures and Tables

**Figure 1 cancers-14-03027-f001:**
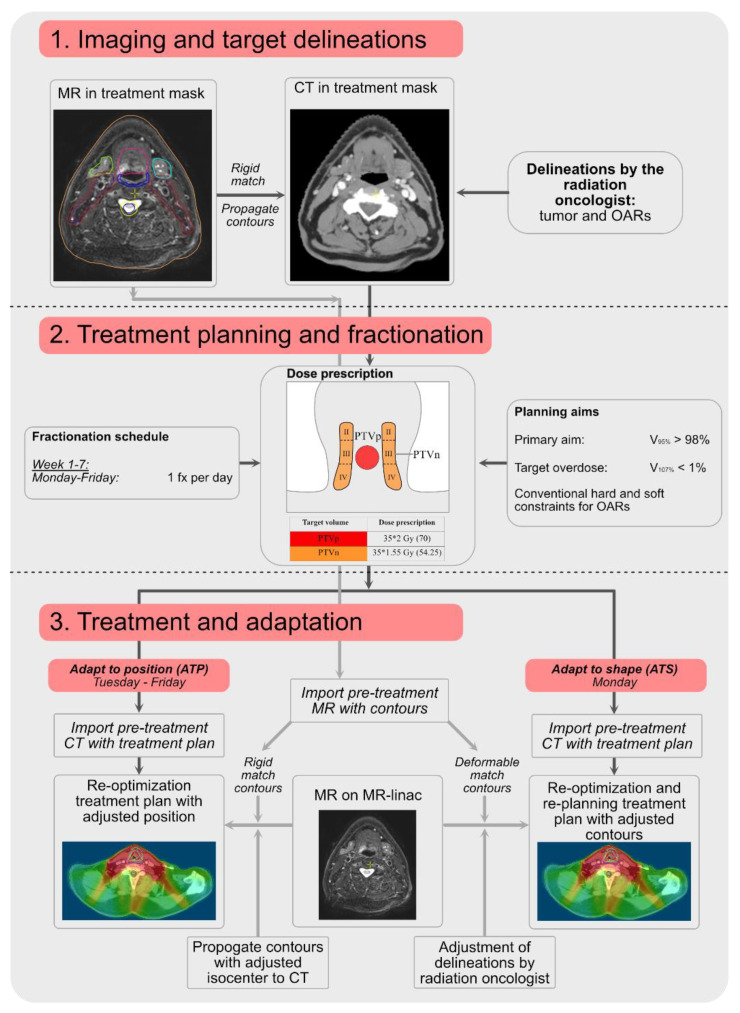
An overview of an adaptive workflow of an HNSCC treatment on an MR-Linac using both adapt to position (ATP) and adapt to shape (ATS). The black arrows point out the trajectory for the CT and the grey arrow for the MRI. Delineations (step 1) are performed by the radiation oncologist (tumor) and radiation therapist (RTT) (OARs). Step 2 is performed by RTT. For step 3, there is a difference between an ATP procedure (fully performed by RTT) and an ATS procedure (registration: RTT; delineation: radiation oncologist; treatment planning: RTT). The medical physicist is supervisor of RTT for planning and registration (on call). This is very user and country dependent.

**Figure 2 cancers-14-03027-f002:**
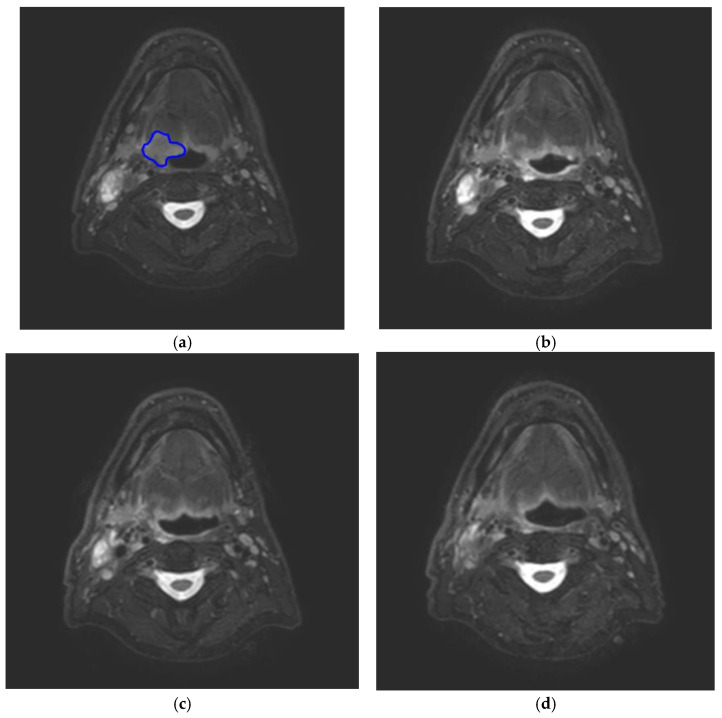
Four consecutive T2 TSE mDIXON images of a patient with a T2N1 HPV+ oropharyngeal carcinoma originating in the base of tongue on the right side (blue) treated with radiotherapy showing the shrinkage of the tumor. Week 1 (**a**), week 2 (**b**), week 3 (**c**) and week 4 (**d**).

**Figure 3 cancers-14-03027-f003:**
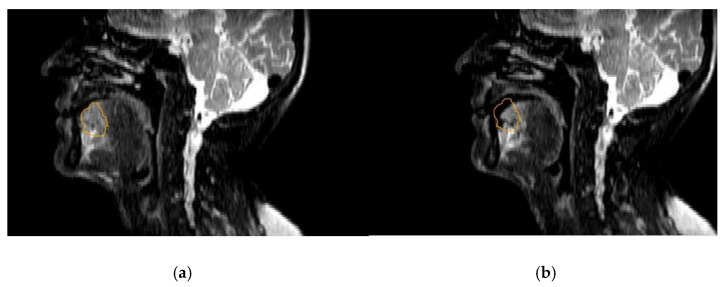
A patient with a metastasis in the tongue that was treated with 6 × 6 Gy on an MR-Linac. The picture showing the difference in GTV position (orange) on T2w 3D TSE SPAIR pre (**a**) (left) and post (**b**) treatment (interval between scans 23 min).

**Figure 4 cancers-14-03027-f004:**
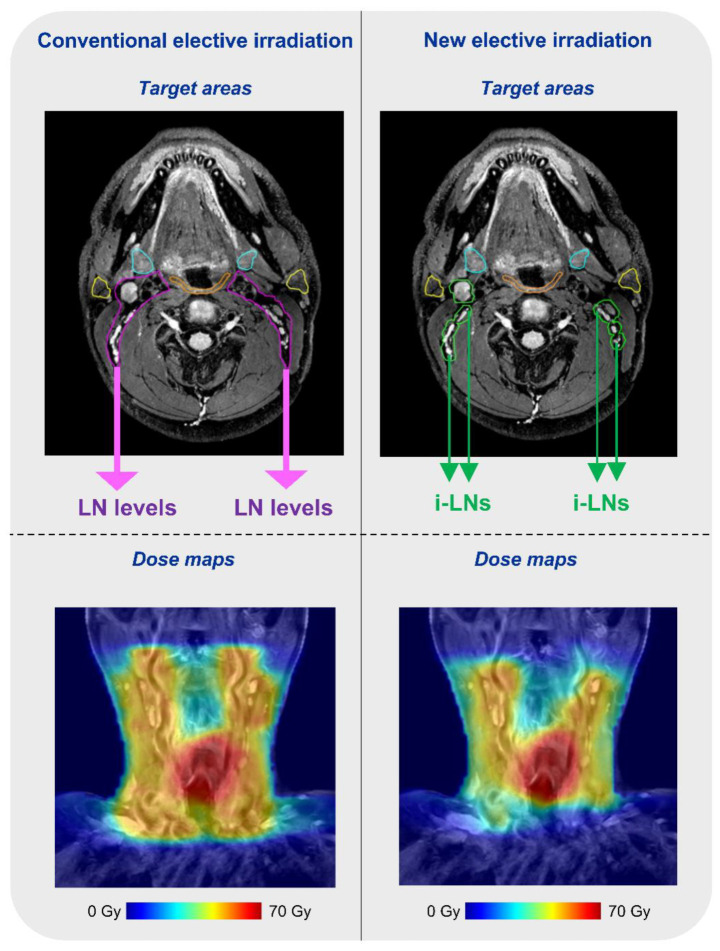
Lymph node-level (**upper left**) and individual lymph node (**upper right**) delineation of a head and neck cancer patient with corresponding dose distributions (**below**).

**Table 1 cancers-14-03027-t001:** Overview of acquisition times per sequence for a head and neck (pharynx) MRI protocol.

MR Sequence	Acquisition Time (Minutes)
M Survey	00:31
B0 map calibrate	01:55
T T2 TSE mDIXON AP	06:39
T DWI SPLICE RL	06:13
S T1 FFE cine	00:59
T T1 TSE RL	02:39
Dynamic13 RL	01:29
T T1 3D TFE mDIXON gd	04:47
Total protocol	25:10

Abbreviations: TSE: turbo spin echo; DWI: diffusion-weighted image; FFE: fast field echo; TFE: turbo field echo; gd: gadolinium.

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
