# Peer review of "Target Definition in MR-Guided Adaptive Radiotherapy for Head and Neck Cancer"

_cancers, 2022, doi:10.3390/cancers14123027_

Round 1

Reviewer 1 Report

Referee report for: Target definition in MR-guided adaptive radiotherapy for head and neck cancer

Brief summary

The authors present a review of recent improvements and potential future developments in MR guided radiotherapy for head and neck cancer. They focus primarily on the MR linac technology and how this changed / will change the treatment workflow.

General concept comments

Qualitative / quantitative review

The authors report numerous studies and imaging techniques. Their discussion is comprehensive and clear. However, always qualitative. The only quantitative data reported is the time needed for ATS and ATP workflows in lines 345-348. Numerical values of GTV-CTV-PTV margins were never reported, but discussed. It would be centrally interesting for the reader having an overview of the different contributions to the margins from inter-observer variability, imaging technique, immobilization, and how much novel techniques can reduce these margins.

Proton discussion missing

Proton and ion beam therapy are cited several times, but never discussed. The technology is rapidly evolving and MR guided proton and ion therapy is under investigation. In this context, additional techniques may be available to detect changes and trigger adaption (eg PET and prompt gamma monitoring). I would recommend to include a sub-chapter dedicated to this topic.

Time discussion missing

The authors present several potential advantages from MR imaging thanks to novel sequences. However, in clinical routine, every additional sequence used for treatment planning increase significantly the time the patient spends in the scanner. This is particularly relevant in adaptive RT, where such MR scans are repeated multiple times over the course of treatment. An overview of scanning times and discussion towards applicability in clinical routine is fundamental and missing.

Paragraph 2.4

The authors present the MR-only workflow. They report the need of sCT. However, a full MR-only workflow requires the use of an MR-linac for the delivery of the fractions and daily-positioning with MR imaging. The sole substitution of CT with sCT while performing treatments at a conventional linac still requires kV or CBCT imaging for daily positioning. This introduces additional challenges since bony anatomy is generally poor quality in sCT but is however critical for patient positioning at a conventional linac with kV or CBCT. A discussion on this topic is missing and necessary.

Specific comments

Line 24.

“MGRT” was not defined. Did the author mean “MRgRT”?

Line 26.

Daily adaption compensates for inter-fraction but not intra-fraction changes. The latter may be mitigated with gating and tracking.

Lines 79-82.

I would recommend to include references to proton and ion range uncertainties.

Figure 1.

The figure includes black and grey arrows and links. Please specify the difference. I would also be interesting for the reader to have an overview of which actions require an oncologist, a physicist or a therapist.

Line 94.

Since the authors discuss both photon and proton treatments, not only the electron density is relevant but also the stopping power.

Lines 185-194.

The authors discuss novel PET techniques within the sub-chapter “The new role of multiparametric MRI”. Could it be an option moving the discussion to the sub-chapter dedicated to CT or PET-CT?

Line 234.

The reference [66] is certainly relevant. However, the seminal paper for sCT generation with neural network is Han Med Phys . 2017 Apr;44(4):1408-1419

Line 238.

A more recent review can provide a better overview to the reader of the available sCT methods for H&N: Spadea et al Med Phys.2021;48:6537–6566-

Line 253.

The results in [67] were confirmed by a prospective study performed on an MR linac: van Timmeren et al Cancers 2021, 13(21), 5404

Line 386.

Should the title be “reducing the target volume”?

Author Response

Please find attached our reply to your comments

Reviewer 2 Report

Well organized and written article. 

Author Response

Thank you

Reviewer 3 Report

This paper is a literature review on the topic of MRI-guided radiotherapy. The authors have comprehensively presented the topic related to the current state of knowledge on adaptive target identification using MR during HNSCC radiotherapy. The paper contains carefully selected and thoroughly analyzed literature. Congratulations on a job well done

Author Response

Thank you

Round 2

Reviewer 1 Report

Thank you very much for addressing my comments. The manuscript is mature for publication.

Please only correct the typo in Table 1 for the second time entry.